# Historical Context Changes Pathways of Parental Influence on Reproduction: An Empirical Test from 20th-Century Sweden

**Cristina Moya** [1,2,*] **, Anna Goodman** [3] **, Ilona Koupil** [4] **and Rebecca Sear** [4]

1 Department of Anthropology, University of California, Davis, CA 95616, USA
2 Centre for Culture and Evolution, Brunel University London, Uxbridge UB8 3PH, UK
3 Department of Population Health, London School of Hygiene and Tropical Medicine, London WC1E 7HT, UK; anna.goodman@lshtm.ac.uk
4 Centre for Health Equity Studies (CHESS), Stockholm University, 106 91 Stockholm, Sweden; ilona.koupil@su.se (I.K.); rebecca.sear@lshtm.ac.uk (R.S.)
* Correspondence: moya@ucdavis.edu

**Abstract:** Several studies have found that parental absences in childhood are associated with individuals' reproductive strategies later in life. However, these associations vary across populations and the reasons for this heterogeneity remain debated. In this paper, we examine the diversity of parental associations in three ways. First, we test whether different kinds of parental availability in childhood and adolescence are associated with women's and men's ages at first birth using the intergenerational and longitudinal Uppsala Birth Cohort Study (UBCoS) dataset from Sweden. This cultural context provides a strong test of the hypothesis that parents influence life history strategies given that robust social safety nets may buffer parental absences. Second, we examine whether investments in education help explain why early parental presence is associated with delayed ages at first birth in many post-industrial societies, given that parents often support educational achievement. Third, we compare parental associations with reproductive timing across two adjacent generations in Sweden. This historical contrast allows us to control for many sources of heterogeneity while examining whether changing educational access and norms across the 20th-century change the magnitude and pathways of parental influence. We find that parental absences tend to be associated with earlier first births, and more reliably so for women. Many of these associations are partially mediated by university attendance. However, we also find important differences across cohorts. For example, the associations with paternal death become similar for sons and daughters in the more recent cohort. One possible explanation for this finding is that fathers start influencing sons and daughters more similarly. Our results illustrate that historical changes within a population can quickly shift how family affects life history.

**Keywords:** fertility; reproductive timing; family structure; life history strategies; educational attainment; cohort effects

## 1. Introduction

Parental absences in an individual's early life are often associated with earlier reproductive development both physiologically (Matchock and Susman 2006; Surbey 1990; Sheppard and Sear 2012) and behaviorally (McLanahan and Bumpass 1988; Chisholm et al. 2005). However, a recent review suggests this phenomenon is not consistent across populations, particularly when considering associations beyond those of fathers' absences on daughters' development (Sear et al. 2019).

These empirical observations present two puzzles for evolutionary social scientists. First, why should fathers be so commonly associated with later reproduction for daughters? Natural selection favors strategies that increase reproductive success. Parents delaying reproduction contradicts one straightforward evolutionary prediction that help from family should improve individuals' fitness, in part by allowing them to reproduce earlier, and

therefore more often. The human data also counters several observations of delayed or suboptimal reproductive behavior among non-human mammals experimentally reared without parents (Wuensch 1985; Bastian et al. 2003; Schradin and Pillay 2004). Second, how can we account for heterogeneity in the associations between parental absence and reproduction across populations? That is, which ecological and cultural factors moderate the role parents play in their children's life histories and reproductive careers?

Evolutionists have proposed several (not necessarily mutually exclusive) theoretical accounts suggesting causal effects between parental absences in childhood and faster reproductive strategies. These include roles for (1) inbreeding avoidance mechanisms (Matchock and Susman 2006); (2) intergenerational conflicts (Moya and Sear 2014); (3) parental guarding of children's reproductive value (Flinn 1988); (4) strategies for extracting parental investments for longer (Ellis 2004); (5) adaptive responses to one's own increased morbiditiy or mortality risks given the lack of parental investments (Geronimus 1991; Rickard et al. 2014); or (6) mechanisms for assessing environmental factors, be they mortality risks (Chisholm 1993), unpredictability (Del Giudice 2014), or the availability of investing partners (Thornton and Camburn 1987; Draper and Harpending 1982). Some of these accounts make further predictions about the circumstances under which parental effects on children should be strongest. For example, the intergenerational conflict model predicts that parental effects should be smaller for sons than daughters when paternity uncertainty is high (Moya and Sear 2014), and some life history accounts suggest parental delays on reproduction only make sense for intermediately harsh environments (Coall and Chisholm 2003).

In this article we focus on a form of the parental investment model; specifically, we examine whether parental presences delay reproduction in part because parents support and encourage education, which in turn trades off with early reproduction. Parental deaths in childhood are often associated with negative educational outcomes (Case and Ardington 2006; Willführ 2009; Gertler et al. 2004). There are several ways parents may influence educational outcomes. For example, parents may directly invest in their children's primary and secondary schooling, thus improving the socio-economic returns to university education for their children. In other words, students who are well-prepared for university, perhaps in part due to parental support, stand to gain more from continuing their education rather than starting to work. Alternatively, parents may directly support their children engaging in status-seeking behavior, which includes higher education in many societies. In this paper we do not differentiate these forms of cultural capital that parents may provide (Bourdieu and Passeron 1990).

The fact that higher education is a historically novel form of social status that can tradeoff with reproduction, may help explain some of the puzzling parental delays to reproduction in post-industrial contexts. It is not surprising that parents commonly have an interest in their children attaining culturally relevant forms of status. However, they also have an interest in encouraging pro-natal behavior, even in low fertility societies (Newson et al. 2005). Often, and arguably for most of human evolutionary history, these goals were not at odds. Culturally defined status is often associated with indicators of individuals' fitness—i.e., long-term number of descendants (Cronk 1991; Borgerhoff Mulder 1987; Hopcroft 2006; von Rueden et al. 2011). However, this relationship between reproductive success and indicators of socio-economic status has weakened, and sometimes reversed, in many post-demographic transition and transitioning contexts (Huber et al. 2010; Snopkowski and Kaplan 2014; Goodman and Koupil 2009; Beydoun 2001; Upchurch and McCarthy 1990; Goodman et al. 2012; Kaplan et al. 1995; see Stulp and Barrett 2016 for a recent review). Cultural evolutionary forces may recently—but perhaps also in societies such as ancient Rome (Caldwell 2004)—favor forms of status that are not necessarily associated with fitness (Boyd and Richerson 1985; Colleran 2016). This means a historical analysis is necessary for understanding how people adopt new forms of status and how the role of parents in shaping fertility behavior changes in the process.

Associations between indicators of status and reproductive outcomes are complex, however; they often show notable sex differences, can be non-monotonic, depend on which

indicators of socio-economic position are used, and may change over time (Sobotka et al. 2017). A consistent finding across contexts is that education, for example, is negatively associated with fertility for women, who experience more tradeoffs between reproduction and education or market labor. In contrast, in some post-demographic transitions contexts, education may be positively associated with reproductive success for men, largely because it reduces the likelihood of childlessness (Kravdal and Rindfuss 2008; Jalovaara et al. 2019; Nisén et al. 2018). Similarly gendered-relationships between fertility and academic success (Goodman and Koupil 2010) and cognitive skills (Kolk and Barclay 2018) have been documented in Sweden. However, the strength of these associations may differ between contexts and change over time. As higher education becomes increasingly common, particularly for women; there is evidence that the negative gradient between education and fertility is nearly disappearing for women in some low fertility settings, including Sweden (Dribe and Smith 2020; Jalovaara et al. 2019). Nevertheless, higher levels of education are consistently associated with delayed first births for women (Neels et al. 2017), and often for men (Corijn and Klijzing 2001; but see Trimarchi and Van Bavel 2017 for discussion of inconsistent associations between education and first births), given cultural norms and economic constraints reduce the likelihood of childbearing until education is complete. There is some evidence that the association between education and age at first birth timing is also becoming more similar for men and women in more recent Swedish cohorts (Dribe and Stanfors 2009).

Despite the vast demographic literature on parental effects on reproduction and on historical changes in reproductive and educational strategies, little has been done to address the ways the role of parents can also change. Here, we address the following three questions in a longitudinal dataset from 20th-century Sweden:

1.　Are parental deaths or separations in childhood associated with reproductive timing?
2.　Are parental associations with age at first birth mediated via educational attainment, specifically university attendance?
3.　Are there historical changes in the pathways of parental influence? More specifically, do changes in men's and women's university attendance accompany changes in how parental absences are associated with their sons' and daughters' reproductive timing?

In the next section we discuss the state of the literature. Section 3 introduces the dataset, historical and cultural context, and describes how they are uniquely suited to addressing these questions. We then discuss our analytical methods. Section 4 describes the results with respect to each question in turn. In Section 5 we discuss how well the answers to each question and other patterns in the data fit the various theoretical accounts.

## 2. Previous Work

### 2.1. Do Parental Deaths or Separations in Childhood Directly Affect Reproductive Timing?

Parental absences can be associated with both delayed and earlier first births. In some pre-demographic transition societies early parental absences correspond to delayed reproductive events suggesting the importance of parents for coordinating marriages, investing in childcare, provisioning grandchildren, and transmitting cultural knowledge about parenting (Scelza 2010; Lahdenperä et al. 2004; Lahdenperä et al. 2007; Jennings et al. 2012; Waynforth et al. 1998; Allal et al. 2004; Mattison et al. 2014; but see Winking et al. 2011 for a society where father absences have no effect). However, most research on the topic, conducted in post-industrial settings, shows the reverse pattern, namely parental absences in childhood expedite the onset of reproduction (Kiernan 1992; Ellis et al. 2003; Palermo and Peterman 2009; Lee 2001; Manlove et al. 2000; Wu and Schimmele 2003; Sheppard and Sear 2012; Chisholm et al. 2005; McLanahan and Bumpass 1988). The most consistent, and most studied, effects are those of fathers on daughters' reproductive timing. This partly reflects the fact that fathers are more often missing than mothers from children's lives, and the fact that daughters' puberty and ages at first birth are easier to measure than sons'. While father's presences are more consistently associated with later reproduction for daughters in contemporary high-income societies than in historical samples or those from

lower income countries, this obscures much variation within high and low-income settings that has yet to be explained.

Although much of this literature treats parental absences of any kind similarly, it is increasingly clear that the causes of parental absence matter and likely reflect different pathways of influence. For example, in rural Bangladesh daughters with separated fathers reproduced earlier than those with fathers present, while daughters with fathers who had died or were working as labor migrants elsewhere reproduced later (Shenk et al. 2013). This diversity of parent absent effects suggests different causal pathways of influence. Few studies, however, have clearly demonstrated causal associations between parental absences and children's reproductive outcomes. As parents' deaths and separations are not randomly distributed, selection effects limit our ability to make strong causal claims. For example, some authors have suggested that associations between parental absences and reproductive development may be due to genetic effects that affect both certain behavioral traits, such as likelihood of divorce, and children's physiological development (Barbaro et al. 2016). However, some case studies of parental separation during war serve as pseudo-experiments that strongly suggest a direct causal effect of parental absences in expediting reproduction (Pesonen et al. 2008). Further, the diversity of associations suggests that environmental context matters, minimally making the genetic confounding a plausible mechanism only in some environments (Uchiyama et al. 2021). Here, we exploit a longitudinal dataset which allows us to assess whether such associations change over a period in which significant genetic change is implausible. This would provide further evidence against the suggestion that only genes matter for these associations.

### 2.2. Are Parental Effects on Age at First Birth Mediated via Educational Attainment?

Little work has been conducted to assess whether investments in educational capital mediate the relationship between parental absences in childhood and later reproductive outcomes. In part, this may be because research in this area has focused on physiological pathways of influence that are developmentally earlier than higher educational investments (Hoier 2003; Ellis 2004; Sheppard and Sear 2012). Independent relationships between parental absences and lower educational achievement (Case and Ardington 2006; Gertler et al. 2004), and between education and later ages at first birth (Rindfuss and St. John 1983; Bongaarts et al. 2017) are well documented, particularly in post-demographic transition and transitioning societies. There is indeed good causal evidence that parental divorces and deaths affect educational outcomes, even in late 20th-century Norway where, much like in our study population, robust social safety nets may buffer such family losses (Steele et al. 2009). Furthermore, across all cohorts born in Norway between 1940 and 1964, more educated women, and to a lesser degree men, begin reproducing later in life (Kravdal and Rindfuss 2008). Given these relationships in a similar cultural context, it is plausible that parental absence may result in earlier first births through this educational pathway across 20th-century Sweden.

### 2.3. Are There Historical Changes in the Pathways of Parental Influence?

To our knowledge this is the first paper that addresses whether the pathways of parental influence on age at first births have changed through time. However, several lines of evidence suggest that such changes are plausible.

All our variables of interest have changed markedly over the last 100 years in nearly every country. The 20th century has seen the spread of formal schooling, delays to first births and decrements in family orientation (Newson and Richerson 2009; van de Kaa 1987; Mills et al. 2011; Bongaarts et al. 2017), though the latter two outcomes did not change monotonically, for exampling reversing in many countries post WWII (Sánchez-Barricarte 2018).

Historical changes during the past century could have also changed whether, and how, parents influence their children's reproductive onset. We will consider an indirect path of parental influence through education, and a direct path of influence that collapses all other possible effects. We describe three mechanisms whereby the direct parental effects on

reproductive timing and their indirect effects via education would be expected to decrease through the course of the 20th century, and one mechanism whereby the indirect effect through education might increase or decrease through time. These correspond to the following historical trends; (1) increases in state support for education, (2) increases in non-parental cultural transmission at university, (3) increases in state support for reproduction, and (4) convergence in educational norms between parents and offspring. We discuss each further in turn.

As states have democratized higher education, we might expect that parental support becomes less necessary to attend university, thus reducing the importance of the indirect pathway of parental influence on reproductive onset. We have previously shown that university attendance became both more common and more meritocratic in Sweden across the 20th century (Goodman et al. 2010). Sweden also followed other countries in shifting from a marked male dominance among university attendees to an overall female bias (Goldin et al. 2006). This implies that parental support in educational endeavors and intergenerational inheritance of cultural capital may matter less as state policies can make university attendance achievable for young adults from wider socio-economic and family backgrounds, and that this pattern may be gendered. This leads to the prediction that indirect parental influences on age at first birth via education should go down through the 20th century.

University attendance may independently change direct parental influences on reproductive timing, as it provides new opportunities for cultural transmission from peers and teachers. Several models have shown that the transmission of beliefs from teachers rather than parents facilitate the spread of low fertility behavior that would otherwise be selected against (Cavalli-Sforza and Feldman 1981; Ihara and Feldman 2004; Boyd and Richerson 1985). As peoples' social networks become less kin-based, anti-natal influences may well increase from university peers, mentors, and colleagues (Newson et al. 2005). This leads to an additional prediction that direct parental effects on age at first birth should decrease through the 20th century as university attendance, the importance of peer networks, and therefore non-kin social influences, increase.

The Swedish state has also introduced policies that lower the costs of reproduction and parenting beyond those that democratize university. For example, various expansions of subsidized day care and parental leave in the second half of the 20th century (Hwang and Broberg 1992) may have reduced the importance of alloparental help (i.e., child-rearing aid from non-parents) from grandparents and other kin (though see Schaffnit and Sear 2017 for evidence that emotional, rather than material, support from parents has pro-natal effects in the UK). If adolescents and young adults are relying less on their parents to help rear their infants, this could decrease the direct effects parents have on reproductive decision-making, independent of their educational roles.

The pace of historical change in the 20th century also means that parents and their children may belong to different generational cultures. Because of the extent of social learning that happens between peers, parents and children do not always adopt the same beliefs (Harris 1999; Kline et al. 2013; Moya et al. 2015). This can affect both direct and indirect parental effects. One possibility is that the more similar the parent and child generations' norms, the smaller the normative influence of parents on education and reproduction should be. That is, children need no convincing to behave as their parents wish them to. On the other hand, it is possible that parents can have stronger effects on their children's life history decisions if they share similar norms. For example, an adolescent may be more successful at attending university if their parent agrees with this goal. Historical changes in university attendance can produce intergenerational discrepancies in beliefs about the necessity, propriety, and prestige of higher education, especially for women. As cultural change in beliefs about education slows down there will be less of an intergenerational gap in expectations as the parent and child generations experience more similar social environments. This suggests that, earlier in the 20th century, parents may try to discourage their children's, especially daughters', university attendance given that the generations

have more discrepant life experiences with higher education for women. On the other hand, later in the 20th century, parents may invest more in daughters' education, and thus delay first births more via this indirect pathway.

## 3. Methods

### 3.1. Dataset

To address our questions, we used data from two adjacent cohorts from the Uppsala Birth Cohort Multigenerational Study (UBCoS Multigen). This longitudinal and intergenerational dataset includes 14,192 people born in the Uppsala University Hospital in Sweden between 1915 and 1929, and their descendants, thus spanning most of the 20th century. Of the original cohort, 12,168 still lived in Sweden in the late 1940s and therefore received personal identification numbers that allows them to be linked across national registers. Their children and other descendants were identified through the Multi-Generational Register and could be followed throughout their lives to measure social and biological data of relevance to several health outcomes.

This dataset has various strengths for this investigation: (1) it includes individuals' complete reproductive histories and several socio-economic indicators across generations, (2) it represents a social context where education, healthcare and childcare have long been heavily subsidized by the state (Hoem and Hoem 1996) and (3) it spans a period with little change in fertility rates, but delays to reproduction (Hoem 2005) and a large historical change in higher education attendance, especially for women (Goodman et al. 2010).

The first strength allows us to control for the influence of an unusually large number of potential confounders, including parental and grandparental socio-economic position and parental reproductive history, several of which are known to be intergenerationally transmitted (Borgerhoff Mulder et al. 2009; Murphy and Knudsen 2002).

The two other strengths are of theoretical importance. This cultural context provides a strong test of the hypothesis that parental presence helps explain variation in reproductive and socio-economic outcomes, given that the Swedish state from the early 20th century onwards has provided much social assistance for individuals to attain their educational and reproductive goals (Hoem and Hoem 1996), thus potentially reducing the importance of kin support. On the other hand, 20th-century Sweden represents a context with relatively high rates of paternal care (Duvander et al. 2010), making it more likely that father absence would be important here. The third feature of this dataset is essential for testing our hypotheses about the relevance of historical changes for parental effects on reproductive timing. Shifting rates of university attendance, especially for women, may have consequences for parents' pro-natalism and educational investment strategies for their children. The fact that we can compare two adjacent cohorts, and specifically ones from the same families, allows us to hold many, if not all cultural factors constant.

### 3.2. Samples

We focused our analysis on the second index generation (G2, *n* = 20,727) of the Uppsala Birth Cohort Multigenerational Study (Figure 1a). We refer to the original cohort as the parent generation. The index generation of this multigenerational, longitudinal dataset represents the only cohort for which we have complete information about parental death dates and nearly complete information about their full reproductive lives. Members of the index cohort were born between 1932 and 1990, but 93% of them had reached 45 years of age by the last time they had been observed (see Supplementary Materials Figure S1). This means that to analyze first birth timing we can include the full second generation since the vast majority would have progressed to this event (Figure 1b). We exclude long-term emigrants from Sweden from the analysis because of the difficulty in linking their data. We also examine parental effects on total fertility with a subset of the sample, but these results are presented only in Supplementary Materials Section S5.

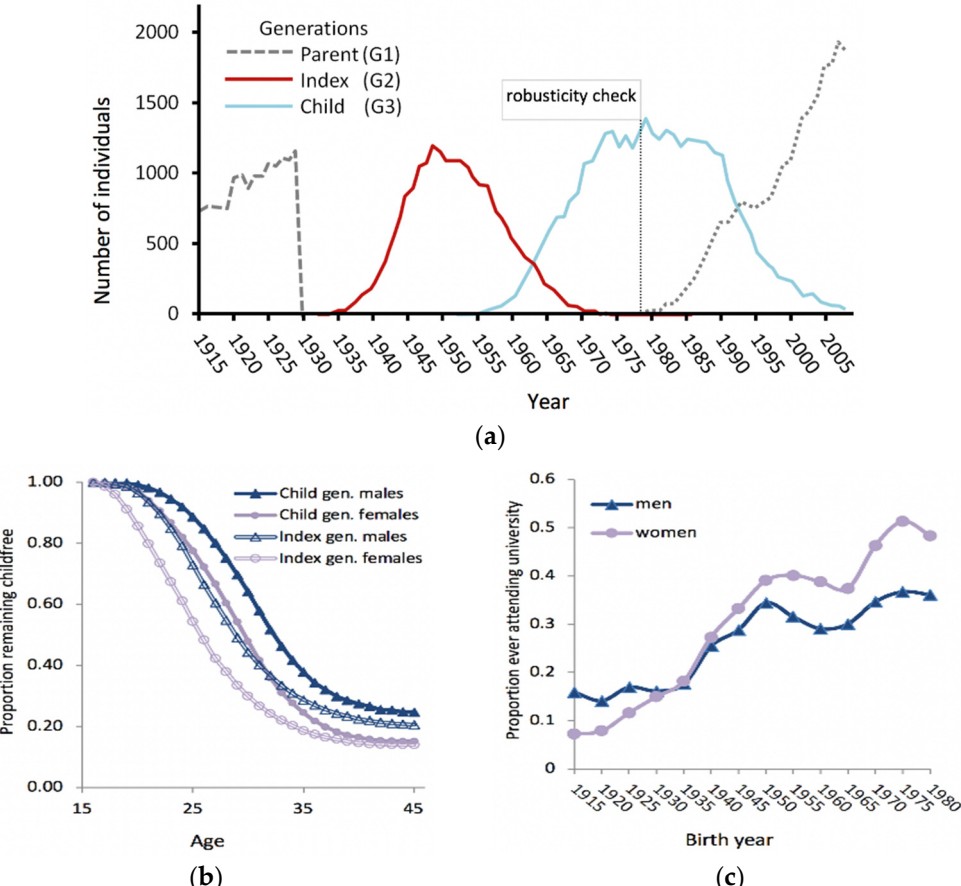

**Figure 1.** Historical changes in UBCoS cohorts. (**a**) Birth years by generation. The red solid line represents the index generation, the primary focus of this study. The child generation in solid light blue was used to test for historical changes in parental effects on first births. Their children are represented by the most recent dotted line. Robusticity checks of individuals older than 35 years of age when last seen would include individuals born before the vertical dotted line. (**b**) Survival curves for progression to first birth. The index generation is shown with open markers, and the child generation with filled in ones. Individuals who were censored for non-reproductive reasons were taken out of the analysis. (**c**) Proportion of adults ever attending university by end of 2009. Plotted by sex and earliest birth year in 5-year birth bands.

To test whether family background associations with timing of first births and their pathways have changed in recent history, we also analyzed ages of first birth for the children of index individuals. We will call them the child generation (*n* = 37,118). There are more individuals in the child cohort who had yet to reach their first birth than in the index generation—i.e., they are censored since they have no age at first birth data. However, event history analyses can incorporate these censored individuals to provide less biased estimates (Singer and Willett 1993). While we use linear regressions that drop childfree people from the analysis in the main text, we find substantively similar results when using event history analyses (see Supplementary Materials Section 2 for robusticity checks).

Several historical changes between the index and child generations are likely to interact with parental effects on timing of first births. Of primary interest to us, (1) there is less intergenerational discrepancy between parents and children in university attendance rates in the more recent generation (Goodman et al. 2010). The gender ratio of university attendees is also more similar for the index and child generation than it is between the parent and index generation (Figure 1c, Table 1). These patterns reflect national level dynamics (Figure S2). Additionally, (2) the median age at first birth is delayed in the child cohort relative to the index cohort (Figure 1b), and (3) through time parental deaths in childhood and adolescence becomes rarer while having unmarried or separated parents becomes more common (Table 1).

**Table 1.** Descriptive statistics for all variables by cohort and gender. For clarity, standard deviations are omitted for categorical variables and means refer to proportion of individuals within that category. NB: many in the child generation had not completed reproduction.

| | | Index (G2) | | | | | | Child (G3) | | | | | |
|---|---|---|---|---|---|---|---|---|---|---|---|---|---|
| | | Men | | | Women | | | Men | | | Women | | |
| | | Mean | SD | *n* | Mean | SD | *n* | Mean | SD | *n* | Mean | SD | *n* |
| Age at first birth * | | 27.33 | 5.52 | 7935 | 24.76 | 5.31 | 8157 | 27.77 | 4.49 | 5864 | 25.98 | 4.54 | 6941 |
| Fertility * | | 2.33 | 1 | 7935 | 2.28 | 0.94 | 8157 | 1.88 | 0.84 | 5864 | 1.99 | 0.89 | 6941 |
| University attendance | | 0.31 | - | 7904 | 0.36 | - | 8143 | 0.29 | - | 5843 | 0.40 | - | 6893 |
| Parental status | (total) | | | 7935 | | | 8157 | | | 5864 | | | 6941 |
| married and cohabiting | | 0.82 | - | 6507 | 0.82 | - | 6678 | 0.61 | - | 3587 | 0.60 | - | 4193 |
| mother dead | | 0.02 | - | 159 | 0.02 | - | 163 | 0.01 | - | 50 | 0.01 | - | 80 |
| father dead | | 0.04 | - | 317 | 0.04 | - | 291 | 0.02 | - | 137 | 0.02 | - | 159 |
| separated | | 0.12 | - | 952 | 0.13 | - | 1025 | 0.36 | - | 2090 | 0.36 | - | 2509 |
| Birth year | (total) | | | 7935 | | | 8157 | | | 5864 | | | 6941 |
| 1930–39 | | 0.02 | - | 159 | 0.02 | - | 177 | - | - | - | - | - | - |
| 1940–44 | | 0.1 | - | 794 | 0.11 | - | 892 | - | - | - | - | - | - |
| 1945–49 | | 0.26 | - | 2063 | 0.25 | - | 2046 | - | - | - | - | - | - |
| 1950–54 | | 0.27 | - | 2142 | 0.27 | - | 2175 | <0.01 | - | 6 | <0.01 | - | 4 |
| 1955–56 | | 0.2 | - | 1587 | 0.20 | - | 1655 | 0.02 | - | 98 | 0.01 | - | 71 |
| 1960–64 | | 0.1 | - | 794 | 0.10 | - | 848 | 0.08 | - | 494 | 0.07 | - | 503 |
| 1965–69 ** | | 0.04 | - | 317 | 0.04 | - | 364 | 0.21 | - | 1220 | 0.20 | - | 1371 |
| 1970–74 | | - | - | - | - | - | - | 0.32 | - | 1879 | 0.30 | - | 2085 |
| 1975-79 | | - | - | - | - | - | - | 0.25 | - | 1490 | 0.26 | - | 1828 |
| 1980–84 | | - | - | - | - | - | - | 0.10 | - | 587 | 0.13 | - | 877 |
| 1985–89 | | - | - | - | - | - | - | 0.02 | - | 89 | 0.03 | - | 188 |
| 1990–94 | | - | - | - | - | - | - | <0.01 | - | 1 | 0.00 | - | 14 |
| Parent's education | (total) | | | 7935 | | | 8157 | | | 5864 | | | 6941 |
| elementary, ≤8 years | | 0.40 | - | 3174 | 0.40 | - | 3283 | 0.05 | - | 312 | 0.05 | - | 353 |
| elementary, 9–10 years | | 0.06 | - | 476 | 0.06 | - | 472 | 0.08 | - | 452 | 0.07 | - | 514 |
| high school, <3 years | | 0.25 | - | 1984 | 0.26 | - | 2102 | 0.38 | - | 2216 | 0.37 | - | 2595 |
| high school, 3 years | | 0.09 | - | 714 | 0.10 | - | 775 | 0.16 | - | 927 | 0.17 | - | 1146 |
| <3 years after high school | | 0.06 | - | 476 | 0.06 | - | 475 | 0.15 | - | 884 | 0.15 | - | 1037 |
| ≥3 years after high school | | 0.11 | - | 873 | 0.10 | - | 855 | 0.17 | - | 990 | 0.17 | - | 1196 |
| post graduate | | 0.02 | - | 159 | 0.02 | - | 145 | 0.01 | - | 82 | 0.01 | - | 98 |
| Parent's household income | | −0.02 | 0.89 | 7935 | −0.04 | 0.86 | 8157 | 0.02 | 0.51 | 5864 | −0.01 | 0.49 | 6941 |
| Grandparent's socioeconomic status | (total) | | | 7935 | | | 8157 | | | | | | |
| higher and mediate non-manual | | 0.1 | - | 794 | 0.10 | - | 791 | - | - | - | - | - | - |
| entrepreneurs and farmers | | 0.21 | - | 1666 | 0.20 | - | 1655 | - | - | - | - | - | - |
| lower non-manual | | 0.07 | - | 555 | 0.07 | - | 582 | - | - | - | - | - | - |
| skilled manual | | 0.15 | - | 1190 | 0.15 | - | 1253 | - | - | - | - | - | - |
| unskilled manual, production | | 0.28 | - | 2222 | 0.29 | - | 2367 | - | - | - | - | - | - |
| service | | 0.19 | - | 1508 | 0.18 | - | 1509 | - | - | - | - | - | - |
| Fertility, mother's | | 2.85 | 1.42 | 7899 | 2.82 | 1.42 | 8133 | 2.52 | 1.05 | 5862 | 2.54 | 1.06 | 6941 |
| Fertility, father's | | 2.91 | 1.46 | 7763 | 2.86 | 1.45 | 7999 | 2.55 | 1.12 | 5827 | 2.53 | 1.10 | 6887 |
| Age first birth, mother's | | 24.34 | 4.41 | 7899 | 24.39 | 4.43 | 8133 | 22.23 | 3.58 | 5862 | 22.34 | 3.61 | 6941 |
| Age first birth, father's | | 27.43 | 4.95 | 7763 | 27.41 | 5.00 | 7999 | 24.69 | 4.01 | 5827 | 24.89 | 4.03 | 6887 |

* does not account for censoring. ** all Cohort 2 members born after 1965 were coded as a 1965–1969 birth band due to small samples thereafter.

Swedish individuals born in the early 1930s represent the first cohort where women would outnumber men among university attendees. The UBCoS parent and index cohorts perfectly straddle this historic shift. All the original cohort members were born before 1930 while all index cohort members were born after 1930. This means that when studying index individuals we are examining a context where parents and children experienced different educational norms (statistical, if not prescriptive norms), but not remarkably different reproductive norms (compare parents' and children's ages at first birth in Table 1). The index generation and their children, on the other hand, belong to cohorts with relatively more similar university gender composition and attendance rates.

*3.3. Variables*

3.3.1. Dependent Variable

All variables used in regression models are described in Table 1. To test hypotheses regarding timing of first births we ran analyses in two ways; first as linear regressions

on age at first birth (AFB), and second as discrete-time event history analyses predicting probability of progressing to a first birth from age 16 onwards given that one had not already done so. The former was used for the mediation analysis.

### 3.3.2. Independent Variables

As our primary predictor we considered any evidence of a parental absence within the first 20 years of life, including deaths and separations. Parents' marital status was available from decadal censuses (from 1960–1990) so we could only get rough estimates of the timing of separations. From this we extracted any evidence of parental separation or non-cohabitation during the first 20 years of one's life. This includes either parent being single, divorced, married but not living with the child's parent, or cohabiting with someone other than the child's parent. While we will refer to these as parental separations, this will admittedly capture many different kinds of families, including cohabiting parents who are not married, a family form that has increased in Sweden since the 1960s (Goodman and Koupil 2009). We construct a single categorical variable of parental availability; parents alive and married and cohabiting, mother dead, father dead, and both parents alive but separated. There were too few individuals whose parents both died in the first 20 years of their life to be analyzed so these were excluded. There are roughly twice as many people who experienced paternal deaths compared to maternal deaths in our sample, and separations are an order of magnitude more common than any parental death. This means that focusing on significance will mislead readers about the importance of each kind of associations. That is, we are more likely to find significant effects of parental separations than parental deaths all else equal, simply because of sample size differences. To avoid such misrepresentation, we focus our comparison of results on effect sizes rather than statistical significance.

### 3.3.3. Mediator

To examine whether parental effects on reproductive outcomes are mediated by their investments in children's education we considered university attendance as a mediator. Given that university, rather than primary or secondary, education most directly trades off with onset of reproduction in post-industrial societies, and years of schooling is very left skewed, we used a binary variable of having ever attended university.

### 3.3.4. Covariates

Given the importance of shared environmental and genetic effects between parents and children (Pettay et al. 2005), we included as many family background controls as possible to reduce confounding. We use three variables to account for family socio-economic background: grandparents' occupation, parents' income, and parents' education. The first is a 6-level categorical variable where grandparents labor is coded as higher and mediate non-manual, entrepreneurial/farming, lower non-manual, skilled manual, unskilled manual in production, and unskilled manual in service. This variable is only available for the index (G2) generation's grandparents. Parent's income reflects the average of parents' disposable income during their adulthood (ages 21–65) standardized by calendar year, age and gender. Income data was combined from the Longitudinal Database for Education, Income and Occupation (1990–2008), in addition to decadal censuses from 1960 onwards. This is modeled with a linear and squared term to allow for non-linear effects. Finally, the maximum educational level achieved by a parent is a categorical variable with 7 levels of schooling; 8 years of elementary or less, 9–10 years of elementary, fewer than 3 years of high school, 3 years of high school, fewer than 3 years beyond high school, at least 3 years beyond high school, and postgraduate schooling. We us two variables to account for family reproductive background; parents' fertility and ages at first birth. Parents' fertility was proxied by the mother's when this data was available, and the father's number of children otherwise. Similarly, the mother's age at first birth was included in models. For the few individuals for whom this data was unavailable, we estimated it by using the father's age

at first birth minus the cohort-specific mean age discrepancy between male and female ages at first birth. Additionally, to account for historical changes within each cohort we adjusted for birth bands of roughly 5-year intervals. For further details about the construction of variables see (Goodman et al. 2012).

*3.4. Analysis*

All models were run in Stata v12 or 13. Stata scripts of the main models are provided in the Supplementary Materials Section 3. For ease of interpretation, we ran models on female and male participants separately given their systematic difference in timing of first birth and educational attainment. When model predictions are shown, average marginal effects were calculated (Bartus 2005) and confidence intervals around the predicted values were estimated using the delta method.

Linear regressions are used to model age at first births, and event history analyses for progression to first birth are provided in the supplement to account for uneven censoring across cohorts. Given the non-independence of observations within a family we use robust standard errors, clustered by shared mother, or shared father if mother identity was missing.

To address the role of higher education as a mediating factor, we model this binary variable with a logit model, employing the user-written command, binary_mediation, to do so. All the same covariates are included in the mediation analyses, but we did not cluster by shared parent. We report bootstrapped percentile confidence intervals (using 500 replications) since sampling distributions of indirect effects tend to be skewed (Preacher and Hayes 2008).

## 4. Results

*4.1. Are Parental Deaths or Separations in Childhood Associated with Reproductive Timing?*

Focusing on the index generation for now, all forms of parental absences in the first 20 years of life are associated with earlier first births for sons and daughters (Table 2, Figure 2). While many of these relationships are not significant, even the smallest among them amounted to decreasing the individual's predicted age at first birth by a quarter of a year. The associations are generally larger for daughters than for sons. For example, a father's death and parental separations are associated with an age at first birth that is nearly 1 year earlier for daughters. It is only a mother's death which is more strongly associated with lower ages of first birth for sons than daughters'—although the association was not significant in either sex. The patterns are substantively the same when only analyzing individuals who reached 35 years of age (Figure S3). Event history analysis of progression to first births generally replicates this pattern of sons' progressions to first births being more strongly associated with maternal deaths (but also separations), while daughters' tempo is more strongly associated with paternal deaths and separations (Figure S4).

*4.2. Are Parental Effects on Age at First Birth Mediated via Education?*

Parental absences in the first two decades of life are generally associated with lower likelihood of attending university (Figure 3, Table S1). Focusing on the index cohort (G2) for now, parental separations are associated with the largest, and most reliable, decrements in the log odds of university attendance. The effect sizes of early maternal deaths on daughters' university attendance rival those of parental separation for this generation. Mothers' and fathers' deaths show roughly the same associations with sons' educational attainment, though neither is significant, while fathers' deaths have effectively zero association with a daughter's university attendance in the index generation.

**Table 2.** Linear regression models predicting age at first birth from parental presence within first 20 years of life. Robust standard errors controlling for family clusters (parent's id) are given.

| | Index (G2) | | | | | | Child (G3) | | | | | |
|---|---|---|---|---|---|---|---|---|---|---|---|---|
| | Men | | | Women | | | Men | | | Women | | |
| | B | SE | p | B | SE | p | B | SE | p | B | SE | p |
| **Parental status (ref = married and cohabiting)** | | | | | | | | | | | | |
| mother dead | −0.6 | 0.47 | 0.2 | −0.44 | 0.4 | 0.27 | 0.43 | 0.51 | 0.4 | −0.83 | 0.52 | 0.11 |
| father dead | −0.26 | 0.31 | 0.4 | −0.96 | 0.3 | 0.001 | −0.66 | 0.36 | 0.07 | −0.85 | 0.35 | 0.02 |
| separated | −0.26 | 0.2 | 0.18 | −0.91 | 0.17 | <0.001 | −0.02 | 0.13 | 0.86 | −0.35 | 0.12 | 0.003 |
| **Birth year (ref = 1932–39)** | | | | | | | | | | | | |
| 1940–44 | −0.52 | 0.46 | 0.26 | 0.01 | 0.35 | 0.99 | - | - | - | - | - | - |
| 1945–49 | −0.05 | 0.45 | 0.91 | −0.05 | −0.34 | 0.89 | - | - | - | - | - | - |
| 1950–54 | 1.08 | 0.45 | 0.02 | 1.22 | 0.34 | <0.001 | - | - | - | - | - | - |
| 1955–56 | 2.08 | 0.46 | <0.001 | 2.56 | 0.36 | <0.001 | −1.14 | 1.9 | 0.55 | 2.32 | 0.75 | 0.002 |
| 1960–64 | 2.3 | 0.48 | <0.001 | 3.32 | 0.38 | <0.001 | −0.89 | 1.84 | 0.63 | 2.61 | 0.53 | <0.001 |
| 1965–69 ** | 2.07 | 0.53 | <0.001 | 3.3 | 0.43 | <0.001 | −0.4 | 1.83 | 0.83 | 2.63 | 0.52 | <0.001 |
| 1970–74 | - | - | - | - | - | - | −0.24 | 1.83 | 0.9 | 3.38 | 0.52 | <0.001 |
| 1975–79 | - | - | - | - | - | - | −1.75 | 1.83 | 0.34 | 2.56 | 0.52 | <0.001 |
| 1980–84 | - | - | - | - | - | - | −4.6 | 1.83 | 0.01 | 0.19 | 0.52 | 0.71 |
| 1985–89 | - | - | - | - | - | - | −8.23 | 1.85 | <0.001 | −2.88 | 0.53 | <0.001 |
| 1990–94 | - | - | - | - | - | - | −10.72 | 1.86 | <0.001 | −6.12 | 0.68 | <0.001 |
| **Parent's education** | | | | | | | | | | | | |
| elementary, ≤8 years | 0.76 | 0.68 | 0.26 | −0.15 | 0.75 | 0.84 | −4.28 | 0.66 | <0.001 | 0.39 | 0.5 | 0.43 |
| elementary, 9–10 years | 1 | 0.72 | 0.16 | 0.32 | 0.78 | 0.68 | −3.46 | 0.68 | <0.001 | 1.08 | 0.51 | 0.04 |
| high school, <3 years | 1.13 | 0.69 | 0.1 | 0.58 | 0.76 | 0.44 | −3.3 | 0.64 | <0.001 | 1.35 | 0.49 | 0.01 |
| high school, 3 years | 1.63 | 0.7 | 0.02 | 1.45 | 0.77 | 0.06 | −2.94 | 0.66 | <0.001 | 1.89 | 0.51 | <0.001 |
| <3 years after high school | 2.2 | 0.73 | 0.003 | 1.63 | 0.79 | 0.04 | −2.96 | 0.66 | <0.001 | 2.07 | 0.51 | <0.001 |
| ≥3 years after high school | 2.34 | 0.71 | 0.001 | 2.32 | 0.78 | 0.003 | −2.49 | 0.66 | <0.001 | 2.62 | 0.51 | <0.001 |
| post graduate | 2.91 | 0.81 | <0.001 | 3.12 | 0.89 | <0.001 | −1.9 | 0.78 | 0.01 | 2.95 | 0.64 | <0.001 |
| Parents' household income | 0.38 | 0.1 | <0.001 | 0.51 | 0.1 | <0.001 | 1.43 | 0.16 | <0.001 | 1.43 | 0.16 | <0.001 |
| Parents' household income | −0.03 | 0.02 | 0.22 | −0.03 | 0.03 | 0.35 | −0.27 | 0.05 | <0.001 | −0.27 | 0.05 | <0.001 |
| **Grandparent's socioeconomic status (ref = higher & mediate non−manual)** | | | | | | | | | | | | |
| entrepreneurs and farmers | −0.19 | 0.27 | 0.48 | 0.2 | 0.24 | 0.4 | - | - | - | - | - | - |
| lower non-manual | −0.52 | 0.3 | 0.09 | −0.31 | 0.29 | 0.28 | - | - | - | - | - | - |
| skilled manual | −0.73 | 0.27 | 0.01 | −0.27 | 0.25 | 0.28 | - | - | - | - | - | - |
| unskilled manual, production | −0.9 | 0.25 | <0.001 | −0.65 | 0.24 | 0.01 | - | - | - | - | - | - |
| service | −0.55 | 0.27 | 0.04 | −0.63 | 0.25 | 0.01 | - | - | - | - | - | - |
| Parent's fertility | −0.15 | 0.05 | 0.002 | −0.18 | 0.05 | <0.001 | −0.08 | 0.06 | 0.21 | −0.34 | 0.06 | <0.001 |
| Parent's age at first birth | 0.15 | 0.02 | <0.001 | 0.19 | 0.02 | <0.001 | 0.14 | 0.02 | <0.001 | 0.15 | 0.02 | <0.001 |
| Constant | 22.46 | 0.94 | <0.001 | 19.15 | 0.91 | <0.001 | 29.22 | 1.99 | <0.001 | 19.71 | 0.88 | <0.001 |
| n | 7935 | | | 8157 | | | 5864 | | | 6941 | | |

** All Cohort 2 members born after 1965 were coded as a 1965–1969 birth band due to small samples thereafter.

Mediation analyses show that these parental contributions to children's university attendance partly explain associations between parental absences and ages at first birth. This indirect effect via education corresponds to lines a*b in Figure 4, while parents' residual direct effect after accounting for university attendance is represented by line c. See Table S2 for all effects used in mediation analyses, and standardized direct and indirect effects.

In the index generation parental absences expedite first births primarily through direct effects rather than indirectly via education (Table 3, Figure 5). The main exception to this pattern is the effect of parental separations on sons' age at first birth, for which the indirect and direct effects are equally large. While we also find a significant, and similarly sized, indirect effect of parental separation on daughters' age at first birth via education, the overall effect of parental separations on daughters' age at first birth is much larger, and this is driven by their direct effects. The partial mediations result from parental separations being associated with lower probabilities of university attendance, which in turn are associated with earlier first birth.

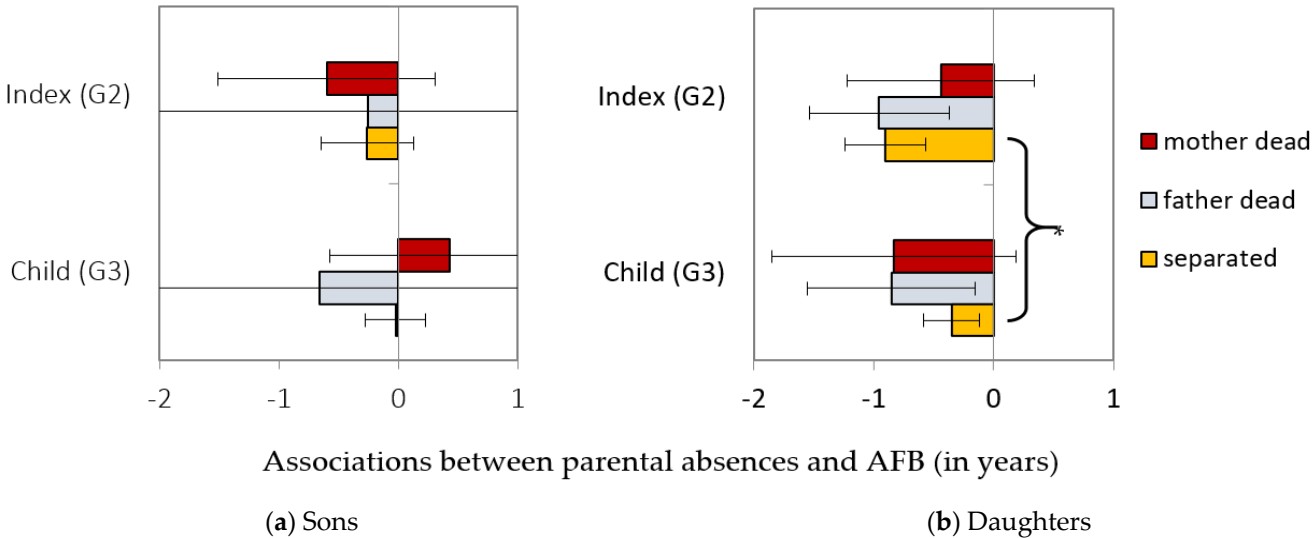

**(a)** Sons                                                    **(b)** Daughters

**Figure 2.** Associations between parental absences and age at first birth (AFB), for (**a**) sons and (**b**) daughters by generation. All associations are shown relative to having married and cohabiting parents and are derived from separate models for each cohort and sex. Tests for differences between cohorts are derived from models with interactions between parental absences and cohort. Robust 95% CIs are shown. * $p < 0.05$.

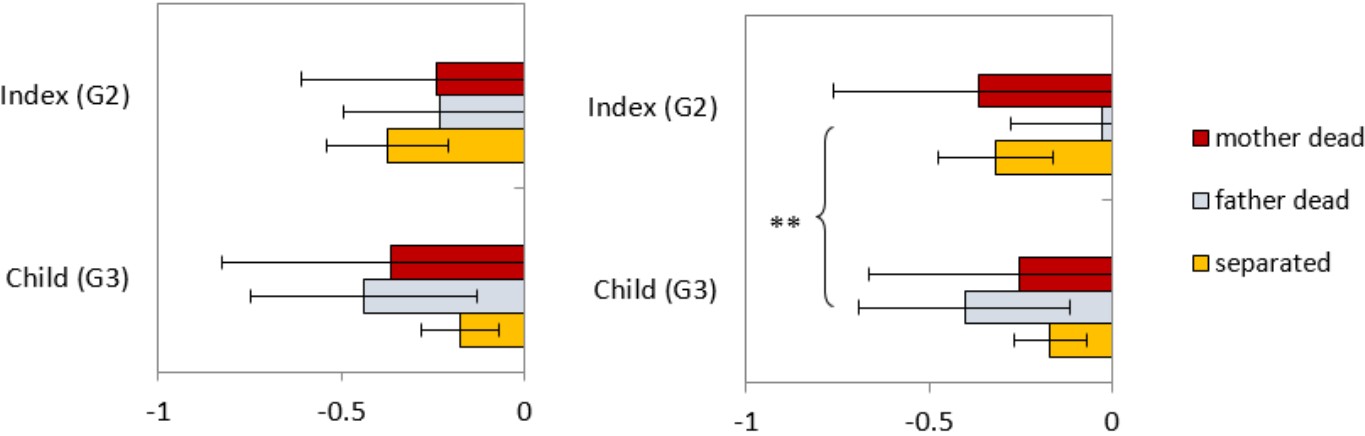

**(a)** Sons                                                    **(b)** Daughters

**Figure 3.** Associations between parental absences and university attendance for (**a**) sons and (**b**) daughters, by generation. All associations are shown relative to having married and cohabiting parents and are derived from separate models for each cohort and sex. Tests between cohorts are derived from models with interactions between parental absences and cohort. Robust 95% CIs are shown. ** $p < 0.01$.

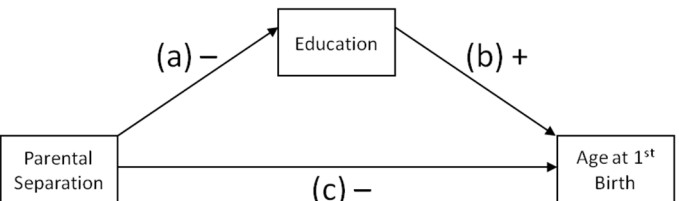

**Figure 4.** Schematic of mediation analyses run for each measure of parental absence. The direction of effects is accurate for most models. Deviations from these directions are specified in Figure 5.

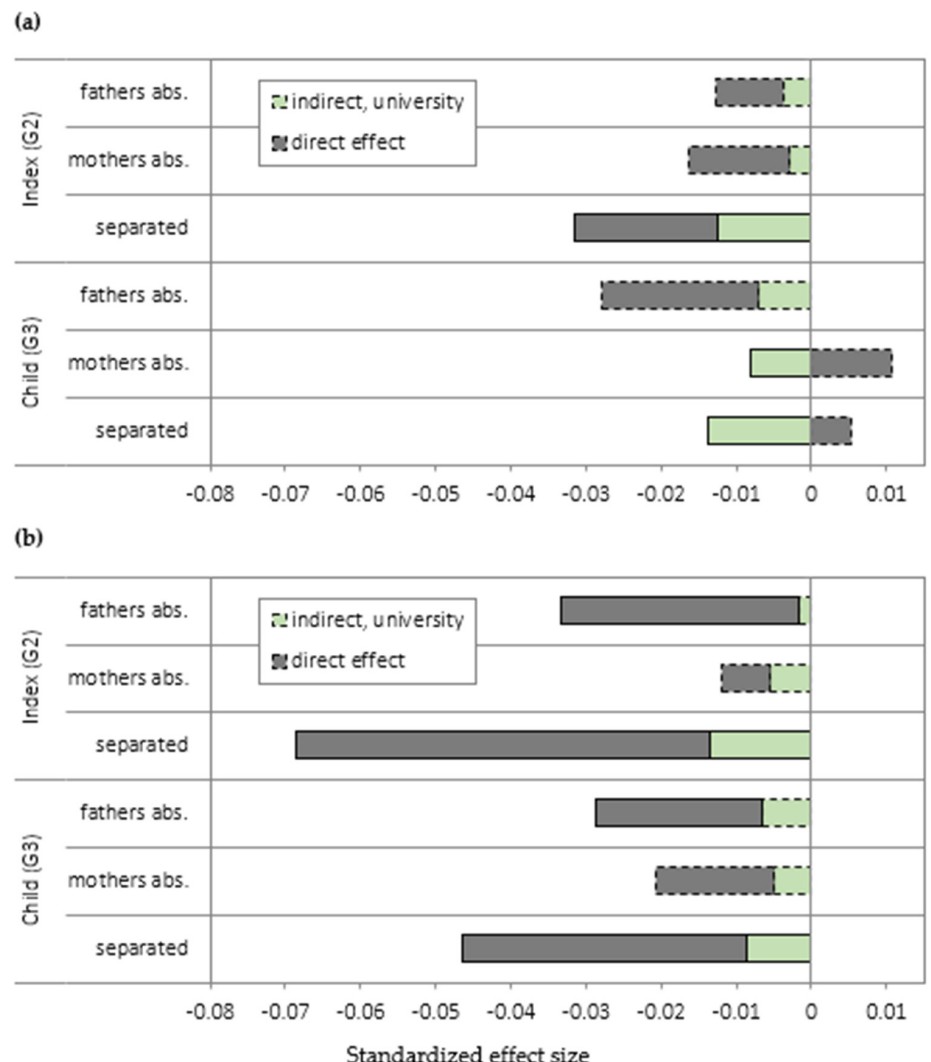

**Figure 5.** Standardized Direct and Indirect effects of parental absences on age at first birth for (**a**) sons and (**b**) daughters, by generation. Effects are relative to having married and cohabiting parents. University attendance is a mediator. Only effects with solid outlines have bootstrapped percentile 95% confidence intervals that did not include zero. NB: the directions of effects deviate from the Figure 4 schema in that for sons in the *child* generation (G3) maternal deaths and parental separations have positive direct effects on age at first birth (c is +).

**Table 3.** Standardized effects of mediation analysis by gender, generation, and type of parental absence. Bootstrapped standard errors are shown.

| | | Sons | | | | Daughters | | | |
|---|---|---|---|---|---|---|---|---|---|
| | | Index (G2) | | Index (G2) | | Cohort 2 | | Child (G3) | |
| | | Coef | SE | Coef | SE | Coef | SE | Coef | SE |
| separations | indirect | −0.013 | 0.003 | −0.014 | 0.004 | −0.014 | 0.003 | −0.009 | 0.004 |
| | direct effect | −0.019 | 0.011 | 0.005 | 0.013 | −0.055 | 0.009 | −0.038 | 0.012 |
| | total effect | −0.031 | 0.012 | −0.009 | 0.014 | −0.068 | 0.010 | −0.046 | 0.012 |
| mother death | indirect | −0.003 | 0.003 | −0.008 | 0.005 | −0.006 | 0.004 | −0.005 | 0.004 |
| | direct effect | −0.014 | 0.010 | 0.011 | 0.011 | −0.006 | 0.011 | −0.016 | 0.012 |
| | total effect | −0.016 | 0.010 | 0.003 | 0.011 | −0.012 | 0.011 | −0.021 | 0.013 |
| father deaths | indirect | −0.004 | 0.003 | −0.007 | 0.004 | −0.002 | 0.003 | −0.007 | 0.004 |
| | direct effect | −0.009 | 0.011 | −0.021 | 0.012 | −0.032 | 0.010 | −0.022 | 0.012 |
| | total effect | −0.013 | 0.011 | −0.028 | 0.013 | −0.033 | 0.011 | −0.029 | 0.012 |

*4.3. Are There Historical Changes in the Pathways of Parental Influence?*

Compared to the index generation, most parental absences in the child generation are similarly associated with earlier reproduction (Figure 2). The only statistically significant difference between the generations is that the association between parental separation and daughters' age at first birth becomes weaker. In other words, there is a significant interaction between generation and parental separation (B(SE) = 0.44(0.2), *p* = 0.03) in the daughter model. However, several interactions with generation are even larger, though not significant. These include the association between a mother's death and a daughter's age at first birth becoming stronger (B(SE) = −0.53(0.65), *p* = 0.41), the association between a mother's deaths and sons' reproductive timing reversing (B(SE) = 0.88(.70), *p* = 0.20), and the strength of the association between a father's death and son's age at first birth increasing (B(SE) = −0.60(0.46), *p* = 0.19). This suggests that our analysis may be underpowered to detect historical changes in the consequences of parental deaths given that these are so rare. Examining progressions to first birth to deal with censored individuals in the more recent cohort reveals similar patterns (Figure S4).

All kinds of parental absences are associated with lower chances of university attendance in the more recent child generation (Figure 3). This stands in contrast to the lack of an association between a father's death and daughter's university attendance in the index generation. This is the only significant interaction between cohort and parental absence, i.e., a father's death is more deleterious to a daughter's higher educational prospect in the more recent generation (B(SE) = −0.51(0.19), *p* = 0.008). No other interaction between generation is of comparable size. Father presence also becomes a significant predictor of son's university attendance in the more recent generation reflecting the second largest interaction with generation B(SE) = −0.29(0.2), *p* = 0.15). While paternal deaths become more negatively associated with university attendance through time, the associations with parental separations become more muted for both sons and daughters through time, though not significantly so.

There are several qualitative differences between the pathways of parental influence on age at first birth in the two generations. Unlike in the index generation, fathers' deaths have very similar effects on daughters and sons in the more recent child generation (Figure 5). While the direct effects of a father's death on age at first birth in the child generation continue to play a larger role than indirect effects, their indirect negative effects via university attendance increase to similar levels for daughters and sons alike. Any suggestion that parents began influencing sons and daughters in more similar ways is tempered by the fact that maternal deaths show more different patterns for sons and daughters in the more recent child generation. Finally, while historical changes influenced the indirect pathway of father effects, the direct pathways of influence shifted more for parental separations. That is, the more muted associations between parental separation and age at first birth in the more recent generation are not due to changes in the ways they affect educational attainment. For sons this means that in the more recent child generation the association between parental separations and timing of their first births is completely mediated by their effects on his university attendance with no remaining direct effect. For daughters the direct effect of parental separations continues having a large, but smaller, expediting effect.

It is worth noting that the two generations' samples are slightly different for two reasons. First, the more recent generation's reproduction is censored. That is, individuals in the child generation who have reproduced are likely to represent young reproducers. Second, we did not have access to the 2000 census from which we derived parental separation measures, meaning that more recently born people might be incorrectly coded as having married and cohabiting parents when in fact a separation ensued by 2000. We conducted a sensitivity analysis restricting the mediation models to individuals who were at least 35 years of age when last seen, meaning that they are more likely to have undergone a first birth, and would have been at least 20 by the 1990 census. Figure S5 shows the very similar patterns of direct and indirect effects for this restricted sample.

## 5. Discussion

We show that parental absences in childhood have gender-specific relationships with life history strategies, that some of the effects are partially mediated by parental effects on university attendance, and that these pathways of influence change across the 20th century along with increases in university attendance, particularly by women. The changing patterns suggest that intergenerational genetic correlations or unmeasured socio-economic variance are unlikely to be the sole explanation for associations between childhood parental absences and reproductive outcomes (Surbey 1998; Comings et al. 2002).

### 5.1. Parental Absences Expedite First Births

The data suggest that parental absences expedite first births, though supplementary analyses suggest they have relatively little effect on total fertility. While several life history theorists have predicted such effects on reproductive timing rather than fertility (Quinlan 2007; Ellis 2004), it is possible that reproductive timing effects are simply more likely to be detected in low fertility populations.

If we had only focused on effects in the index generation, we might have concluded that opposite-sex parental deaths have larger effects on age at first birth than same-sex parental absences. However, this pattern does not hold in the child generation. In the more recent generation, the associations between mothers' deaths and daughters' ages at first birth are equivalent in size to those of fathers' deaths, even if only the latter are significant. Similarly, in the more recent generation paternal deaths are more strongly associated with earlier ages at first birth for sons than maternal deaths are. Most of the previous literature has focused on effects of father, rather than mother, absence, and some studies comparing fathers and mothers directly have suggested stronger developmental acceleration effects of father absence (Bogaert 2005). We only partially confirm this trend for daughters in the index generation and sons in the child generation. Our study illustrates the need to focus on effect sizes rather than significance for such claims given that some associations with maternal and paternal absences are of similar effect sizes but differ in their statistical significance in part because of the larger sample of dead fathers than mothers. This greater incidence of paternal than maternal deaths is typical of human datasets given the lower life expectancies and later ages of reproduction for men than women in nearly all societies.

Similarly, the literature has primarily focused on female children and adolescents' development for a series of methodological, policy, and theoretical reasons—menarche is easier to measure than adrenarche, women's teen pregnancies are seen as a larger social problem (Card and Wise 1978), and the tradeoffs between early reproduction, and somatic and educational investments may be starker for women (Ellis 2004). However, direct comparisons are rare and generally there seems to be few clear patterns regarding the interaction of parent and child sex on developmental outcomes (Russell and Saebel 1997). Our study shows that the developmental consequences of family disruptions on age at first birth tend to be larger for daughters, but can be as severe for sons as they are for daughters (e.g., mothers' deaths was more strongly associated with sons' than daughters' ages at first birth in the index generation).

Parental separations are more reliably associated with daughters' first births than sons' (Figure 2). Unlike some previous work (Biblarz and Gottainer 2000; Kiernan 1992; Shenk et al. 2013), we find that the associations between age at first birth and parental separations are not notably larger than those with parents' deaths. In fact, in the index generation the effect sizes of parental separations and father deaths are nearly the same, while the more recent child generation the effect of associations with father deaths are stronger. This pattern in the index generation may reflect that separations traditionally meant less interaction with fathers than with mothers, thus paralleling the effects of paternal death. On the other hand, the associations between parental separations and ages at first birth are much smaller in the child generation. This suggests that the social context has a large effect on the meaning of parental separations. In the recent generation being an unmarried or separated parent was less likely to be stigmatized, represented less

self-selected families, may have been less of a financial burden for the primary caretaker, or may have resulted in relatively more equal parenting by mothers and fathers.

### 5.2. Parental Associations with Age at First Birth Are Only Partially Mediated by University Attendance

Most parental absences in childhood are associated with lower university attendance rates (Figure 3, Table S1), which in turn reduces age at first birth. These indirect effects are relatively small compared to the direct effects of parental absence on age at first birth, especially for daughters (Figure 5). Nonetheless, we can discern reliable indirect effects of parental separations via their effects on university attendance for both sons and daughters in both generations. Notably, paternal deaths are not associated with a daughters' university attendance in the earlier generation (Figure 3b), and thus have no indirect effect on their age at first birth. Those indirect pathways which are observed via education are most consistent with theoretical proposals that children delay their maturation when they get higher parental investments and can reap these benefits (Ellis 2004). The fact that fathers' absences were not detrimental to women's higher education in the mid twentieth century precludes such an indirect pathway of influence and may reflect the differential reproductive opportunity costs of university attendance for men and women (Dribe and Stanfors 2009). Arguments could also be made that the indirect pathway of influence through education is also consistent with hypotheses that adolescents speed up reproduction in parent absent contexts that either affect, or are associated with, higher mortality, morbidity, or few investing partners (Geronimus 1991; Draper and Harpending 1982; Chisholm 1993) if forgoing higher education is part of that strategy.

The generally larger direct pathways of parental influence may reflect a variety of mechanisms. For example, parents can improve their children's socio-economic success in ways not captured by university attendance. Conversely, parental absences in the first 20 years of life reflect continuing absences into adulthood in the case of death, and likely lower social support and investment in the case of separations. These different life trajectories may mean that people with investing parents can delay reproduction due to lower weathering (Geronimus 1991; Rickard et al. 2014) or benefit more from investing in non-reproductive skills (Ellis 2004). In line, with the idea that parents guard their children's reproductive value (Flinn 1988), the direct pathway may reflect parenting strategies that prevent their children from engaging in non-normative or stigmatized behavior. While this account was developed to explain parents' guarding of daughters' behavior, teenage fathering may also pose unwanted social and economic risks to sons that parents would be keen to prevent.

While the direct pathway could reflect mechanisms proposed by other theories, the results and cultural context give us reason to be skeptical of their importance. If inbreeding avoidance mechanisms are at play (Matchock and Susman 2006), we would have expected opposite-sex parental absence associations with their child's age at first birth. This pattern only holds for the earlier cohort. Intergenerational reproductive conflicts are relatively unlikely in this society where children do not particularly contribute to household production or childcare, providing little material incentive for parents to keep their children at home (Moya and Sear 2014). Finally, using parental absences as a cue to mortality risks (Chisholm 1993) or unpredictable environments (Del Giudice 2014) should result in similar effect sizes for parents and children of any gender, while using absences as a cue to the availability of investing fathers (Draper and Harpending 1982) would only predict father absence effects. The complicated patterns of variation across parent and child sex and across generations make it likely that multiple mechanisms are involved.

### 5.3. Pathways of Parental Influence Change across Time

If parental effects simply changed to reflect intergenerational changes in reproductive norms, parents would delay their children's first births more in the more recent generation, given that the mean age at first birth increased for both men and women across these cohorts. While we do not find this pattern for all parent-offspring dyads, we do see it for

father-son and mother-daughter associations. That is, fathers are more strongly associated with later first births for their sons and mothers are more strongly associated with later first births for their daughters in the more recent child generation compared to the previous generation, though not significantly so. This may reflect the fact that children are also motivated to follow the reproductive norms of their generation leaving little room for parental social influence.

If the expansion of governmental programs facilitating reproduction changed parental effects on reproductive decision-making, we would have expected parental effects to become weaker through time. More specifically, parents should have become less necessary to carry out reproduction resulting in less expediting effects of parents. It is difficult to assess whether this mechanism is at play in the historical changes we see given that having parents around does not generally expedite reproduction. This means that we see little evidence that young adults are using their parents as alloparental resources that help them reproduce earlier. The only pattern consistent with this is that maternal deaths delay their sons' reproduction, but only in the more recent generation and not significantly so. If anything, this pattern argues against the hypothesis that more generous parenting policies in recent generations weakened parental effects.

Nor do we find that governmental policies supporting higher education reduced the effect of parental absences on university attendance. The only kind of parental absence that had a smaller effect on education in the more recent cohort was that of parental separations. We believe it is more plausible that this is due to changes in the meaning, stigma, and economic consequences of parents' marital status through time, than a reflection of policy changes improving educational access. In fact, the effect of parental separations on son's age at first birth is completely mediated by the effect separations have on education in the more recent cohort, suggesting that pathways via other psychological consequences are of decreasing importance.

In contrast we find that fathers' deaths are more strongly associated with lower university attendance in the more recent generation, though the cohort difference is only significant for daughters. This translates into larger relationships between fathers' deaths and age at first birth for sons only, while for daughters it changes the fathers' pathway of influence, but not his total effect on their ages at first birth. The fact that the direct and indirect effects of paternal deaths look similar for sons and daughters in the more recent generation suggests that changing gender norms around university attendance affect how fathers interact with, and invest in, their children. In other words, the more recent fathers, who themselves belonged to a cohort with greater female than male university attendance, may have supported relatively similar educational and reproductive timing strategies for both sons and daughters. In contrast the index generations' fathers, who belonged to a cohort with markedly higher university attendance for men than women, were only associated with increased university attendance for their sons. This interpretation does not explain why mothers had similar effects on their sons' and daughters' education across the generations. However, some recent evidence suggests that parents experience greater conflict over how to rear adolescent girls than boys and that this is driven by fathers having gender-role attitudes that are discrepant with their daughters' (Kabátek and Ribar 2020). The larger paternal effects on education overall may also reflect a greater involvement of fathers in child-rearing in the more recent generation.

Finally, if peer and non-kin social networks were influencing reproductive timing more as university attendance increased, we would expect to have seen decreasing direct effects of parents through time. This is only the case for opposite-sex parents and parental separations. As discussed previously, we believe the decreasing direct effects of parental separations are better understood considering the changing meaning, selection into, and social consequences of reproducing outside of marriage. The direct effects of mothers on sons are nearly equivalently sized but in opposite directions across cohorts, and those of fathers on daughters are nearly the same as their effects on sons in the child cohort.

This suggests that waning parental social influence might not account for these historical changes in pathways of effect either.

Consistency over time can also be of interest if it implies that more canalized mechanisms (i.e., those less susceptible to environmental changes) are at work. Interestingly, the reduction in the direct effect of parental separations across time seems to have been larger for sons than daughters. On the other hand, the indirect effects of parental separations on reproductive onset via university attendance are consistent across time for both daughters and sons. In both generations daughters' reproductive timing is more susceptible to the effects of separations, suggesting that at least some of the mechanisms at work are robust to changing and more egalitarian gender norms.

## 6. Conclusions

We present evidence that parental absences in the first 20 years of life are associated with reproductive timing—even using a rather weak approximation of presence and investment (by their vital or marital status), and even in a society that provides much governmental support for reproduction, education, and material well-being. We add to the growing literature showing that parental deaths and separations in the first two decades of a child's life are associated with earlier reproductive outcomes in post- demographic transition societies. Associations between parental absences and age at first reproduction are partly explained by their influence on university attendance that delays first births, but residual direct effects tend to be larger. Furthermore, we see that the historical context moderates the total effects, and pathways of influence, of parental absences. The associations with fathers' death become more similar for daughters and sons in the more recent generation suggesting the importance of gendered norms about higher education in how parents influence children. Fathers' effects via education may also have increased through time due to greater male parental involvement, but if so, the change is less pronounced for sons. Parental separations are less strongly associated with earlier reproduction in the more recent generation, suggesting that the social consequences of raising children outside of marriage are becoming less severe, and perhaps even normative. However, the fact that daughters' ages at first birth are more strongly associated with parental separations than sons' ages at first birth is robust to historical changes. This suggests that the pathways of developmental influence differ depending on the kind of parental absence tested, and that some are more likely to change with shifting gender and family norms.

Our results indicate some promising directions for future research. The fact that we find different patterns between two adjacent cohorts that share broad cultural similarities suggests caution is warranted in generalizing too widely across societies. For example, if parental separations have different effects across our cohorts because of their changing prevalence or social meaning, it would behoove researchers to understand how concepts about separation and selection into these categories differs cross-culturally. Perhaps more problematically for generalizing from this study to the rest of the world, 20th Sweden represents an atypical society in many ways. Particularly relevant to evolutionary and cross-cultural scholars, most societies through time have relied less on biparental care given that alloparents tend to invest heavily in child rearing (Hrdy 2009). Of course, kin and formal institutions such as daycare and schools still play important alloparental role in post-industrial societies like Sweden, but the nuclear family is usually a residential unit and bears a larger share of the childcare burden. In contrast, some non-industrial societies even have specific institutions like partible paternity or levirate for buffering paternal absences (Walker et al. 2015). This may help explain the more reliable effects of father absences in societies that are structured around the nuclear family (Sear et al. 2019). Further cross-cultural work will help us gage the extent to which our results generalize. It should also help us adjudicate between theoretical accounts. At present the literature on parental influences on reproduction offers multiple theoretical explanations, but relatively few attempts at selecting between them. By focusing on the mechanisms of parental influence

and choosing study populations for their theoretical relevance we can make sense of a large empirical literature.

**Supplementary Materials:** Supplementary Tables, Figures and Analyses are available online at https://www.mdpi.com/article/10.3390/socsci10070260/s1.

**Author Contributions:** Conceptualization, C.M., A.G. and R.S.; methodology, C.M. and R.S.; software, C.M. and A.G.; validation, C.M., A.G.; formal analysis, C.M.; investigation, I.K.; resources, I.K.; data curation, A.G. and I.K.; writing—original draft preparation, C.M. and R.S.; writing—review and editing, C.M., A.G., I.K. and R.S.; visualization, C.M.; supervision, I.K. and R.S.; project administration, R.S. and I.K.; funding acquisition, R.S. and I.K. All authors have read and agreed to the published version of the manuscript.

**Funding:** This research was supported by the European Research Council, Starting Grant No. 263760 (PI Sear) and the Swedish Research Council for Health, Working Life and Welfare, FORTE project No. 2018-00211 (PI Koupil).

**Institutional Review Board Statement:** The study was approved by the Regional Ethics board in Stockholm.

**Informed Consent Statement:** Informed consent was obtained from individual participants in the study.

**Data Availability Statement:** To secure the privacy of participants, limited access to the UBCoS dataset is granted to collaborators https://www.chess.su.se/ubcosmg/ (accessed on 2 July 2021).

**Acknowledgments:** We would like to thank Amy Heshmati for help running scripts remotely. We also appreciate the detailed and constructive reviewer comments.

**Conflicts of Interest:** The authors declare no conflict of interest.

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
