# Peer review of "Historical Context Changes Pathways of Parental Influence on Reproduction: An Empirical Test from 20th-Century Sweden"

_socsci, doi:10.3390/socsci10070260_

Round 1
Reviewer 1 Report
Overall, the manuscript is well-written and easy to follow the concept flow. However, the authors need to provide mode detail in the method and result sections.
- For the international readers who are not familiar with Swedish, a introduction of general SES. might be helpful, such as the % of citizen having university education in general. In my country, for example, almost half of citizen has a bachelor degree. Therefore, the university attendance might not be an important indicator here.
- The introduction of dataset is needed to provide more information of sample recruitment, IRB, and so on.
- How the authors ran the analyses? What software program is used? What are the model fits? More details for the analyses and the tables for overall results are needed.
- Edit error: (1) line 337; (2) line 378, can’t find supplementary materials; (3) line 484
- Most citations are out of date. Citing recent articles (e.g., published within 5 years) are needed.
Reviewer 2 Report
This paper uses the Uppsala Birth Cohort Study from Sweden to find out whether different kinds of parental absence in childhood and adolescence are associated with the timing of women’s and men’s first birth and lifetime fertility. They find that parental absences in childhood and adolescence are associated with a lower age of first birth (mostly for daughters not sons). They also determine whether parental presence delays age at first birth indirectly due to parents encouraging /facilitating university attendance. They find that university attendance does partially explain the link between parental presence and delayed first birth.
Much of this paper is overly wordy and needs to be edited, there are problems with incorrect wording, problems in the sub-heading titles, the supplemental tables were unavailable for me to see as a reviewer, and, as a result, some of the findings reported in the text are mysterious. Despite the abstract promising an analysis of lifetime fertility, this is not presented in the current paper. (Line 436 suggests they are in the supplemental material, but this was unavailable to me). Nonetheless, as an analysis of the effects of parental presence on age at first birth for men and women and the role of university attendance as an intervening/mediating variable it is mostly well done and has some interesting findings.
I do have a number of concerns. First, I dislike the absence of results tables in the body of the paper, as it makes some of the results reported in the text difficult to interpret. While the results tables are in the Supplemental tables that were unavailable to me, I think some of them should also be included in the body of the text. The figures are great, but I would also like to see the full analysis that they are based on.
Second, line 95 the authors write: ” In many post-demographic transition contexts indicators of socio-economic status have become decoupled from reproductive success, particularly for women “ and line 622 “Parental education investments do not necessarily pay off in terms of total reproduction in post-demographic transition settings.” This is true, but it is not true in the Swedish context as the authors surely know. Education and income are negatively associated with number of children for women in Sweden, but for Swedish men, education and income are *positively* associated with number of offspring (Goodman and Koupil 2010; Kolk and Barclay 2018). So parents encouraging education in males in the Swedish context is not encouraging a form of status attainment that is “decoupled from reproductive success.” This should be made clear earlier in the paper.
Third, given the different relationship between education and reproductive behavior for Swedish men and women, the sex differences in the results should be clarified and highlighted more in the paper. The authors write (Line 553) “we show that parental absences in childhood have gender-specific relationships.” The findings reported in the figures show that most of the effect of parental absence on age of first birth is for daughters, not sons. This should be made clear and included in the abstract.
Fourth, the authors should better describe the data set and explain if the data are a probability sample of the Swedish population. If it is a probability sample (or an entire population of a certain age group or cohort), statistical significance is worth noting. Non-significant effects should not be reported as results, unless the data are population data. Of course the confidence intervals are a test of statistical significance in and of themselves. All the control variables should be clearly presented with how they were measured and why they were included. For example, how is occupational status measured? What is a birth band? The Covariates section is far too brief and needs to be expanded (while the rest of the paper needs to be abbreviated).
Specific problems:
Line 28: Authors write: “parental absences tend to be associated with delayed first births”. The authors mean the opposite. They find parental absences tend to be associated with earlier first births (mostly for women).
Line 54 Sentence problem “…parental absence of and reproduction…”
Line 57 Need question mark
Line 154 Heading issue
Line 210 Authors write: “We describe three mechanisms whereby the direct parental effects on reproduction …would be expected to decrease through the course of the 20th century. These three mechanisms should be made clearer. I assume they are: 1. Because of state support for education, parental support may have become less necessary for children to attend university. 2. As university attendance increases, transmission of low-fertility norms could be facilitated 3. State support for families in Sweden could mean a reduction in the influence of parents on their children’s reproduction.
Line 404 Details on covariates and how they were measured should be included here.
Line 492 Without the table, it is hard to interpret this b. Non-significant effects should not be reported as results, unless the data is population data.
Line 553 Authors write: “we show that parental absences in childhood have gender-specific relationships” These should be highlighted more in the paper, especially given the different relationship between education and reproductive behavior for Swedish men and women.
Line 751 References should be organized alphabetically.
References
Goodman, A., & Koupil, I. (2010). The effect of school performance upon marriage and long-term reproductive success in 10,000 Swedish males and females born 1915–1929. Evolution and Human Behavior, 31(6), 425-435.
Kolk, M., & Barclay, K. J. (2018). Cognitive ability and fertility amongst Swedish men. Evidence from 18 cohorts of military conscription. MPIDR Working Paper 2017–020 (Max Planck Institute for Demographic Research, Rostock, Germany). Available at https://www. demogr. mpg. de/papers/working/wp-2017-020. pdf. Accessed February 27.
Round 2
Reviewer 2 Report
This is a much improved paper and a good contribution on the relationship between parental deaths/separations and reproductive timing, and the question of whether educational attainment mediates this relationship.
I realize the references I gave in my last review concerned the relationship between fertility and educational performance, not educational attainment. However, there is evidence of the positive relationship between educational attainment and fertility in Sweden for men. See: Jalovaara et al. (2019), Nisén, et. al 2017 for Finland. See also Dribe and Smith (2020) who find that social class (measured by type of job manual, nonmanual etc) associated with higher fertility for men, and more recently, women, in Sweden.
In the abstract authors write: “our results illustrate that cultural changes within a population can quickly shift how family affect (should be affects) life history.” Clearly, the Swedish context changed over the course of the last century (broadening access to education, greater gender equality in education), but given that you do not have measures of cultural or normative change, your research does not illustrate effects of “cultural change.” Better to use the words you use in the title “historical context”.
Authors write: This includes a nod to the possible role of different reproductive consequences of higher education for men and women. However, it's unclear to us that this difference can account for the full range of gendered effects (e.g. why do mothers encourage daughters education, but not fathers?)
My response: Your results do not show that mothers (but not fathers) encourage daughter’s education. You found father’s death had “effectively zero association with a daughter’s university attendance in the index generation.” You don’t measure “encouragement”, so you do not know what the mechanism is between parental presence and child educational attainment.
Section 4.2 I would like to see the table with the analysis for Figure 5 in the manuscript.
Line 81 The reference to Hopcroft (2002) seems odd, as Hopcroft (2006) is more relevant.
Line 177 “Independent relationships..” Specify relationships here.
- 186 Rephrase last sentence in paragraph
References
Martin Dribe & Christopher D. Smith (2020): Social class and fertility: A long-run analysis of Southern Sweden, 1922–2015, Population Studies
Hopcroft, Rosemary L. (2006). Sex, status and reproductive success in the contemporary U.S. Evolution and Human Behavior 27: 104-120.
Jalovaara, M., Neyer, G., Andersson, G., Dahlberg, J., Dommermuth, L., Fallesen, P., & Lappegård, T. (2019). Education, gender, and cohort fertility in the Nordic countries. European Journal of Population, 35(3), 563-586.
Nisén, J., Martikainen, P., Myrskylä, M., & Silventoinen, K. (2017). Education, other socioeconomic characteristics across the life course, and fertility among Finnish men. European Journal of Population, 1-30.
